# Complex coastlines responding to climate change: do shoreline shapes reflect present forcing or 'remember' the distant past?

Christopher W. Thomas[1], A. Brad Murray[2], Andrew D. Ashton[3], Martin D. Hurst[4], Andrew K. A. P. Barkwith[4], Michael A. Ellis[4]

[1]British Geological Survey, Lyell Centre, Edinburgh, EH14 4AP, Scotland, UK.
[2]Division of Earth and Ocean Sciences, Nicholas School of the Environment and Earth Sciences and Center for Nonlinear and Complex Systems, Duke University, Durham, North Carolina, 27708, USA.
[3]Department of Geology and Geophysics, Woods Hole Oceanographic Institution, Woods Hole, Massachusetts, 02543 USA.
[4]British Geological Survey, Nicker Hill, Keyworth, Nottingham, NG12 5GG, England, UK.

Correspondence to: Christopher W. Thomas (cwt@bgs.ac.uk)

**Abstract.**

A range of planform morphologies emerge along sandy coastlines as a function of offshore wave climate. It has been implicitly assumed that the morphological response time is rapid compared to the time scales of wave-climate change, meaning that coastal morphologies simply reflect the extant wave climate. This assumption has been explored by focussing on the response of two distinctive morphological coastlines - flying spits and cuspate capes – to changing wave climates, using a coastline evolution model. Results indicate that antecedent conditions are important in determining the evolution of morphologies, and that sandy coastlines can demonstrate hysteresis behaviour. In particular, antecedent morphology is particularly important in the evolution of flying spits, with characteristic timescales of morphological adjustment on the order of centuries for large spits. Characteristic timescales vary with the square of aspect ratios of capes and spits; for spits, these timescales are an order of magnitude longer than for capes (centuries vs. decades). When wave climates change more slowly than the relevant characteristic timescales, coastlines are able to adjust in a quasi-equilibrium manner. Our results have important implications for the management of sandy coastlines where decisions may be implicitly and incorrectly based on the assumption that present-day coastlines are in equilibrium with current conditions.

**Keywords**

Capes, spits, characteristic timescales, antecedence, equilibrium, sandy coasts, diffusivity, instability, wave climates

## 1 Introduction

Many recent studies demonstrate how distinctive rhythmic planform coastline shapes on scales of kilometres to hundreds of kilometres (Figure 1) can develop on wave-dominated sandy coasts due to morphodynamic interactions with wave climates (Ashton and Murray, 2006a, b; Ashton et al., 2001; Falqués and Calvete, 2005; Falqués et al., 2000; Kaergaard and Fredsoe, 2013a, b, c; van den Berg et al., 2012; Hurst et al., 2015; Nienhuis et al., 2013). Typically, previous studies assume (implicitly or explicitly) that coastline shapes chiefly reflect the contemporary wave climate (generally with specific regard to the angular distribution of wave angles), adjusting very rapidly to any changes. That is, intrinsic timescales for change in morphology are shorter than the timescales characterizing shifts in wave climate. Such rapid morphological change, relative to the rate of change in the wave climate, reflects a quasi- or dynamic equilibrium morphological response to changing climate forcing: put simply, the rate of change in the morphology is much faster than the rate of change in the wave climate

Changes in storm patterns in future climates will likely yield different wave climates (WASA Group, 1998; Hemer et al., 2013; Storlazzi et al., 2015; Wolf and Woolf, 2006) with concomitant changes in coastline behaviour. Moore, et al. (2013) demonstrated temporal shifts in coastline shape on the Carolina capes caused by observed shifts in wave climate, supporting the prediction that even slight differences in wave climate can be associated with different morphologies (cf. Ashton and

Murray, 2006a, b). Similarly, Allard et al. (2008) related spit growth and morphological change to wave climate variations on the Arçay Spit, France. Changes in wave climates will be manifest in altered patterns of shoreline change, including zones of enhanced erosion and/or accretion (Barkwith et al., 2014; McNamara et al., 2011; Slott et al., 2006). However, the assumption of quasi-equilibrium coastline change, with shoreline shape reflecting the present wave climate, has not been previously examined.

In an analogous study, Nienhuis et al. (2013) have shown how inherited delta morphology can influence the evolution of the coastline when sediment supply is perturbed: the shape of a delta can reflect not just present or recent sediment supply, but can depend strongly on the long-term history of sediment input. Might the influence of former morphology on coastline evolution also be true for sandy coastlines when wave climates are perturbed?

In this paper, we present the results of a study exploring the two key questions that arise from the above summary,

specifically: the influence of former morphology of sandy coastlines on their subsequent evolution as wave climates change with regard to approaching wave angles (focussing on rates and spatial scales of response), and the degree to which coastlines can respond to changing wave climate in a quasi-equilibrium manner.

To address these questions, we have used the one-line Coastline Evolution Model (CEM; Ashton and Murray, 2006a) to examine: a) whether, and under what conditions, coastline shapes exhibit quasi-equilibrium responses to changes in the

angular distribution of approaching waves, and b) whether there are conditions under which coastlines might retain long-term memory of previous wave-climate forcing. Such hysteresis would greatly complicate forecasting and identification of coastline behaviours to be expected with changing wave climate.

We focussed on two end-member coastal morphologies, flying spits and cuspate capes (Figure 1). These morphologies emerge as self-organised structures in response to high angle wave instability that results when waves approaching the coast

at highly oblique angles, causing shoreline undulations to grow (Ashton et al., 2001; Falqués and Calvete, 2005).Their value from an academic perspective notwithstanding, our interest in these particular morphologies extends beyond their fascinating self-organised complexity. These morphologies are themselves important in many coastal regions because they often shelter fragile but highly diverse and dynamic shallow marine and estuarine ecosystems upon which human and animal communities depend and also shelter socially and economically important coastal infrastructure. The fate of capes and spits

under changing climate is thus of material concern to humanity and needs to be better understood to aid the development of appropriately informed coastal management policies.

In our experiments, we generated spit and cape coastlines using appropriate wave angle distributions (Ashton and Murray, 2006a). We then changed the wave angle distributions to be more diffusive in character, such that complex, high amplitude perturbations (as exemplified here by spits and capes) are smoothed. Changes were made either instantaneously or gradually over a period of time. Outputs from the modelling were used to quantif how coastal morphologies responded to these changes.

In this initial investigation, we have focussed specifically on changes in the distributions of approaching wave angle, as these distributions fundamentally control gross coastal morphology at and above kilometre scales. Changes in other wave properties (height, period) control the rates at which changes occur, so are important in coastline evolution. However, we wished only to study the effects of changing wave angle distribution. The decision to move linearly from an unstable to diffusive wave angle distribution was motivated by wishing to understand the degree of stability of complex morphologies under changing wave angle distributions, hence the simplicity of our experiments. More complex experiments involving non-linear or oscillating changes in wave climate were not warranted at this stage, but could be explored in future work. The timescales for change in those experiments where the wave angle distribution was changed gradually were to some extent arbitrary, but guided by timescales over which climate is known to be changing (decades – centuries) under the influence of anthropogenic activity.

## 2 Background and methods

### 2.1 Alongshore gradients and shoreline planform change

Gradients in alongshore sediment flux, generated by wave-driven currents, cause shoreline erosion or progradation on sandy coastlines. Assuming conservation of mass in the shoreface, the temporal change in coastline location in relation to flux gradients is given by:

$$\frac{\partial y}{\partial t} = -\frac{1}{D_{sf}}\frac{\partial Q_s}{\partial x} \tag{1}$$

where $y$ (L) is the cross-shore position at long-shore position $x$ (L) at time $t$, and $Q_s$ (L$^3$ T$^{-1}$) is the alongshore sediment flux; $D_{sf}$ (L) is shoreface depth – that over which erosion or accretion extends (Ashton and Murray, 2006a).

Critically, $Q_s$ is chiefly dependent on the angle between offshore wave crests and the shoreline (Ashton et al., 2001; Ashton and Murray, 2006a; b, figure 2b; Falqués and Calvete, 2005; see Figure 2A), here expressed in terms of the commonly-used CERC equation (Komar, 1971, 1998; Ashton and Murray, 2006a), but re-cast in terms of offshore waves (Ashton and Murray, 2006a):

$$Q_s = K_2 H_0^{\frac{12}{5}} T^{\frac{1}{5}} cos^{\frac{6}{5}}(\emptyset_0 - \theta) \sin(\emptyset_0 - \theta) \tag{2}$$

$K_2$ ($m^{3/5}$ $s^{-6/5}$) is a constant, $H_0$ is *significant* offshore wave height in metres and $T$ is wave period in seconds; $\emptyset_0$ is the offshore wave angle (representing waves at the offshore limit of approximately shore-parallel contours), and $\theta$ the local coastline angle (both in degrees), both relative to some general coastline orientation.

The angle dependency of $Q_s$ is such that sediment flux is maximised at offshore wave approach angles of ~ 45° with respect to the shoreline. Waves approaching from offshore at angles < 45° ('low-angle' waves) will smooth a coastline, such that a straight coastline is in a stable equilibrium. Waves approaching at angles of 45° or greater ('high-angle' waves) induce instability along a coastline and protuberances tend to grow. How coastline morphology evolves depends on the relative degrees of influence of high-angle and low-angle waves in the wave climate, as well as the degree of asymmetry in the wave angle distribution (Ashton and Murray, 2006a).

**2.2 Net Sediment Flux and Shoreline instability and diffusivity**

For a nearly straight coastline, coastline evolution can be described by a linear diffusion equation, where the diffusivity is either positive (stable) or negative (unstable) (Ashton and Murray, 2006a, equation 8). Every approaching wave condition contributes to sediment transport and the consequent evolution of the coastline. The overall effect of a wave climate on a coastline can be determined from the net diffusivity, $\mu_{net}$, (m $s^{-2}$), calculated from the sum of the individual diffusivities induced by each wave condition (analogous to the net alongshore sediment flux), and from a dimensionless 'stability index', $\Gamma$, that measures the competition between stability and instability (Ashton and Murray, 2006b). We use these indices to quantify the behaviour and state of a coastline under the influence of a particular wave climate.

For each wave at each location along a coastline, individual alongshore sediment flux values are calculated using (2), and individual diffusivity ($\mu$) values are calculated with respect to the local coastline orientation using the shoreline diffusivity equation obtained by Ashton and Murray (2006a; b; equation 8):

$$\mu = \frac{K_2}{D} T^{\frac{1}{5}} H_0^{\frac{12}{5}} \left( cos^{\frac{1}{5}}(\emptyset_0 - \theta) \left[ \left( \frac{6}{5} \right) sin^2(\emptyset_0 - \theta) - cos^2(\emptyset_0 - \theta) \right] \right) \tag{3}$$

Summing over individual fluxes and diffusivities at each coastline location over $n$ wave events gives the net flux and diffusivity, respectively:

$$Q_{s,net} = \sum_{i=1}^{n} Q_{s,i} \, \Delta t_i / \sum_{i=1}^{n} \Delta t_i \tag{4}$$

$$\mu_{net} = \sum_{i=1}^{n} \mu_i \, \Delta t_i / \sum_{i=1}^{n} \Delta t_i \tag{5}$$

(Ashton and Murray, 2006b, equation 5). The 'stability index', $\Gamma$, is given by:

$$\Gamma = \sum_{i=1}^{n} \mu_i \, \Delta t_i / \sum_{i=1}^{n} |\mu_i| \, \Delta t_i \tag{6}$$

where $\Delta t_i$ (s) is the time step (Ashton and Murray, 2006b, equation 6).

$\Gamma$ ranges between 1 for a fully low-angle wave climate, and -1 for a fully high-angle climate. Mapping $\Gamma$ along a coastline for different wave climates elucidates the nature of the response of a coastline to those wave climates (cf. Ashton and Murray, 2006b). We have used calculations of $\Gamma$ to quantify the behaviour of coastlines characterized by capes and spits that have

grown under an anti-diffusive wave climate in contrast to those formed under a different, more diffusive wave climate. We subsequently investigated how $\mu$ and $\Gamma$ changed during the transition from anti-diffusive to diffusive wave climates in order to reveal how domains of erosion and deposition would change along a coast with changing wave climate.

### 2.3 Coastal Evolution Modelling

**2.3.1 Coastline planform evolution**

The CEM uses equations (1) and (2) to explore coastline planform behaviour numerically. The details of the theory and its implementation in the CEM are discussed extensively elsewhere (Ashton et al., 2001; Ashton and Murray, 2006a, b), so here we summarize the model domain and setup. In this study, the model domain consisted of $2000 \times 300$ cells alongshore and cross-shore respectively; each cell is $100 \times 100$ m. The model was driven by synthetic offshore wave angle distributions, as

described in Section 3.1.2 below. Model runs used daily time steps, with a new offshore wave angle being chosen at random from the wave angle distribution at each model day.

Wave angle distributions define the relative influences on alongshore transport of all the waves approaching from angles falling within each angle bin (Ashton and Murray, 2006b). Observed (or hindcast) wave records can be transformed into such angular distributions, additionally weighting each wave by height and period ($H_0^{\frac{12}{5}} T^{\frac{1}{5}}$ ; Eq. (2)), and summing all the

wave occurrences in each bin (Ashton and Murray, 2006b). However, in the model experiments reported here, we used simplified synthetic wave distributions. These generic distributions could represent either systematic differences in wave height as a function of approach angle, or in the frequency with which waves approach from different angles (or a combination of the two). The angular distribution of wave influences on alongshore transport determines the emergent dynamic-equilibrium coastline morphology (Ashton and Murray, 2006b); we discuss a relevant measure of dynamic

equilibrium below. Rhythmic coastline features retain a self-similar shape under a constant wave climate, even though the scale of the shape increases slowly through time (Ashton and Murray, 2006a; Ashton et al., 2001). The effective average wave height (Ashton and Murray, 2006b) and period only influence the timescales for coastline development.

Because the principal interest in this study is in the effect of changing wave angle distribution on planform change, the effects of variations in wave period $T$ and height $H_0$ can be folded into angular distributions of wave influences on

alongshore sediment transport (Ashton and Murray, 2006b); in this study, $T$ and $H_0$ were fixed at 10 s and 1 m in all runs.

Using linear wave theory, each offshore wave is refracted progressively over shore-parallel contours until depth-limited breaking occurs (e.g. see Hurst et al., 2015, appendix A). At this point, the standard breaking-wave version of the CERC equation (e.g. Komar, 1998) is used to calculate the sediment flux as a function of the angle between the locally-determined coastline orientation (Figure 2a) and the wave angle at breaking and the breaking wave height. The coastline position is

evolved based on the calculated gradients in flux, assuming conservation of mass in the shoreface (equation 1).

### 2.3.2 Coastline instability and diffusivity

Net flux and diffusivity data generated by the CEM can be used to explore the sensitivity of a coastline to change under existing and modified wave climates, and the processes by which any change would occur, as discussed in Section 2.2. In this study, the CEM was used to capture the *potential* net flux and diffusivity data for particular wave climates operating on a

coastline at particular points in time. To do this, spits and capes were grown for a specific time using wave climates with appropriate *U* and *A* values, as described below in Section 3.1.2. Coastline evolution was then paused and 10,000 sample wave angles drawn randomly from the selected wave climate distribution were run over the extant coastline. The average potential net flux and net diffusivity can then be calculated at each location (cell) along the coastline. This characterizes the *potential* change in the response (either unstable or diffusive) of the coastline to that wave climate, at any point along its

length, and given its current morphological state...

### 3 Experiments with changing wave climate

### 3.1 Experimental design

### 3.1.1 Instantaneous and gradual wave climate change

We set up experiments by growing either flying spits or cuspate capes ('capes' from here-on) from an initially straight coast

(with small white noise perturbations) over an initial fixed period of time. In most experiments, this initial period was 250 model years. This timeframe allows these morphologies to attain length-scales commensurate with those observed along real coastlines. We then subjected these model coastlines either to a gradual change in wave climate, or an instantaneous change. In experiments with gradual wave climate change, the initial wave climate was evolved linearly towards the new state over an arbitrary period of 100 years. These experiments were used to explore the influence of pre-existing morphology on the

nature and rate of response of a coastline to changing wave climate.

In experiments involving instantaneous change, the wave climate is transformed to the target state immediately following the period of initial growth; in these experiments, we also used initial periods of 50 and 125 years to provide additional data that we could use to determine characteristic timescales for change. This allowed us to explore the possibility of scaling relationships between time and the rate of change of length scales, and the degree to which a quasi-equilibrium response in

morphology is possible for given rates of wave climate change.

In both cases, coastline morphology is in dynamic equilibrium with the initial wave climate just before the wave-climate transition begins. As the wave climate changes, the coastline progressively approaches a new morphological state, settling into dynamic equilibrium with the final wave climate. We characterized coastline morphology using the aspect ratio (cross-shore extent/alongshore wavelength) of coastline features: previous work has shown that for equilibrium coastline shapes,

aspect ratio varies with wave climate (Ashton and Murray, 2006a). As a coastline continues to evolve under a constant wave climate, the scale of coastline features grows (Ashton and Murray, 2006a; Ashton et al., 2001). However, aspect ratio

remains constant even as length-scales increase (Ashton and Murray, 2006a). Hence, aspect ratio is an appropriate measure reflecting the degree of dynamic equilibration with respect to a particular wave climate.

### 3.1.2 Experimental wave climates

Model simulations were driven by wave approach angles drawn from a probability density function (pdf) defined by two parameters (Ashton and Murray, 2006a): $U$, the fraction of waves approaching from angles > 45° (representing the fraction of wave influences on alongshore transport from these angles) (Figure 2b) and $A$, the fraction of waves approaching from the left (CEM convention); the wave climate is asymmetric when $A > 0.5$. When $U > 0.5$, the model coastline experiences instability and perturbations will grow. The pdfs used in our modelling are shown in Figure 2b.

Ashton and Murray (Ashton and Murray, 2006a, figure 9) mapped different coastline shapes that emerge for different values of $U$ and $A$. We have explored two trajectories across the $(U, A)$ parameter space in our experiments, one for capes and one for spits (Figure 2c). In both cases, the trajectory is towards a diffusive wave climate, under which perturbations are smoothed.

### 3.1.3 Experimental conditions

Model capes are generated over 250 years with $U$ at 0.7, and $A$ at 0.5 (Figure 3a.i); flying spits are generated with $U$ at 0.7 and with $A$ set to 0.7 (Figure 3b.i). Subsequently, $U$ is changed from 0.7 to 0.55 (moderately anti-diffusive), or to 0.45 (diffusive) while holding $A$ constant (0.5 for capes; 0.7 for spits); $U$ is changed either instantaneously, or gradually over 100 years. The new wave climate is then held constant for a further 650 years. Total model run times are 900 years for models in which the wave climate is changed instantaneously, and 1000 years for models in which the wave climate is changed gradually over 100 years. In addition, to investigate how response times vary with the spatial scale of the features, in separate experiments we generated capes and spits over 50 and 125 years, followed by instantaneous change in $U$ to 0.45

## 4 Results

### 4.1 Changes in morphology under gradual wave climate change

Examples of the changes that occur in the planform morphology of our experimental coastlines during the model runs are shown in Figure 3; note that these data are for experiments in which the wave climate is changed gradually over 100 years from the initial $U = 0.7$ to $U = 0.55$ or 0.45.

In Figures 4 and 5 we plot the evolution of aspect ratio, wavelength and amplitude of coastal features (capes and/or spits) during the experiments. The data have been smoothed using a 7-point moving average window to aid clarity. Animations of the model simulations from which the results discussed in this study were derived are also included in the Supplementary Information.

### 4.1.1 Capes

Cape morphology adjusts mostly in the 100-year period over which the wave climate changes (Figure 4a.i). Cape amplitude declines through erosion of the cape tips. (Figure 3a.ii, first and second panels). Rapidly declining aspect ratio (Mean Amplitude/Mean Wavelength) indicates quasi-equilibrium adjustment in amplitude, approaching that expected for $U = 0.55$ (Figures 3a.ii, second and third panels; 4.Ai). For a diffusive wave climate the coast eventually becomes smooth (Figure 3a.iii, Figure 4a.i,iii).

### 4.1.2 Spits

Flying spits change much more slowly, exhibiting pronounced long-term memory (Figure 4b.i). The coast shape in dynamic equilibrium with $U = 0.55$ features small, relatively low-amplitude reconnected sand-waves (Figure 3b.ii, third panel; cf. Figure 1c; Figure 2c). However, in the experiment in which $U$ changes from 0.7 to 0.55, the coastline morphology differs with regard to both shape and scale from that expected at $U = 0.55$ (Figure 3b.ii, cf. second and third panels), even several centuries after the wave-climate transition ends. More strikingly, when the wave climate becomes diffusive ($U = 0.45$), the experimental coastline retains significant undulations even after 750 years (twice the time it took to grow the flying spits) (cf. Figure 3b.ii, second and third panels). Figure 4b.i-iii shows most of the adjustment occurs after the transition period. The aspect ratio converges toward the dynamic equilibrium value, but only over a timescale of many centuries.

### 4.1.3 Resultant length scales

The results show that for both capes and spits, the scale of both wavelength and amplitude of coastline features is larger than would be expected had the coastlines formed under the final wave climates ($U = 0.55, 0.45$; Figures 3a.ii, b.ii; 4a.ii,iii, b.ii,iii). Under anti-diffusive wave climates, this scale 'memory' is a permanent result of the path through wave-climate space. Shape 'memory', on the other hand, fades over timescales that differ for capes and spits: spits retain a memory of shape over many centuries, capes only over a few decades (Figure 4a.i, bi).

### 4.2 Instantaneous wave climate change: characteristic timescales and temporal-spatial scaling

To derive characteristic timescales for change in morphology and to examine the relationship between temporal and spatial scales, further model experiments were run with initial wave climate conditions lasting 50, 125 and 250 years, followed by an instantaneous change in wave climate. The rates of morphological change in capes and spits are shown in Figures 5a and 5b respectively, with reference to initial conditions run for 250 years.

Using the change in aspect ratio as the metric, we determined characteristic time-scales for morphological change (i.e. first and second e-folding scales, $1/e$ and $1/e^2$) from the point at which the wave climate is changed to become diffusive ($U = 0.45$). For capes, first and second e-folding times are approximately 20 and 80 years (Table S1, Supplementary information), respectively; for spits they are approximately 90 and 320 years (Table S2, Supplementary Information). Comparing characteristic time scales of morphological change in aspect ratio for capes and spits generated over 50, 125 and 250 years,

we find that diffusive scaling pertains, with timescale varying as the square of the characteristic wavelength (Tables S1, S2, Supplementary Information).

## 5. Discussion

### 5.1 Physical Mechanisms of coastline morphological adjustment

Based on the results from the instantaneous change experiments, we can distinguish two modes in which coastlines can adjust to changing wave climate, exemplified by the cape and spit experiments, respectively: Cape morphologies adjust to a zero net flux condition, whilst spit morphologies adjust to a condition in which there is a constant down-drift translation of the feature.

### 5.1.1 Mechanisms of cape morphology adjustment

To understand how capes adjust, we begin by considering capes in dynamic equilibrium with a symmetric wave climate dominated by high-angle waves ($U > 0.5$; Figures 2b, 3a.i). Cape tips feel the full distribution of wave angles in the regional climate. In contrast, cape flanks and inter-cape bays are affected by wave climates that differ from the regional wave climate because of shadowing by adjacent capes (Ashton and Murray, 2006b; Murray and Ashton, 2013). The shadowing shelters the flanks from the higher angle waves in the distribution, the degree of shadowing depending on location in the inter-cape bays.

For a dynamic equilibrium to develop under a symmetric wave climate – a state in which the coastline adjusts very rapidly to small changes in wave climate at short temporal and small spatial scales – all local coastlines must adjust to orientations that produce little or no net sediment flux under the local wave climate. As high-angle waves become more dominant in the wave climate, the local coastlines adjusted to zero net flux lie at progressively greater angles, relative to the regional coastline orientation, towards the cape tips. Consequently, cape amplitudes increase with larger $U$. This behaviour is evident in the

*potential* net flux ($Q_{s,\mathrm{net}}$) and stability index ($\Gamma$) data calculated to explore their spatial variation along a coastline.

Examples of these characteristics are shown in Figure 6a,c for a section of coastline with two sample capes. Figure 6a shows a cape pair from part of a model cape coastline grown for 250 years under a symmetric wave climate with $U = 0.7$. Dynamic equilibrium is indicated by the near-zero net flux condition across most of the inter-cape bay (Figure 6a). In addition, both the net flux and stability index data (Figures 6a, 6c) show that only the cape tips experience the anti-diffusive effects of the

full wave climate, whereas the flanks and the bay experience the diffusive effects of the variably shadowed local wave climate.

As the wave climate shifts to one dominated by waves approaching more directly onshore (e.g. $U = 0.45$; Figures 2b, 3a.ii, iii), orientations previously adjusted to the zero net flux condition become subject to significant change in net sediment fluxes, which are directed away from the cape tips (Figure 6a). The resulting strong gradients in these fluxes cause cape tips

to erode and bays to prograde.  The fluxes are proportional to the maximum net flux divided by the alongshore length scale

for the cape tip (some small fraction of the total cape wavelength). The change is strongly apparent in the stability index data (Figure 6c; blue line). The inter-cape bay is no longer shadowed with respect to the dominant wave directions and becomes relatively less stable under the new wave climate. The very tips of the capes behave more diffusively but stability falls rapidly along the cape flanks.

**5.1.2 Mechanisms of spit morphology adjustment**

For spits, both the mode of emergence of the steady-state pattern and the mode of subsequent adjustment under a changing wave climate are more complicated. For an asymmetric wave climate, under which there is translation of finite-amplitude coastline features (Ashton and Murray, 2006a), a dynamic equilibrium is indicated by constant alongshore translation of the spits.

The tips of spits tend to propagate in the direction parallel to the shoreline orientation that produces the maximum sediment flux for the given regional wave climate (Ashton and Murray, 2006a; Ashton et al., 2016). However, the flanks and tips of spits also experience alongshore translation caused by erosion at their updrift ends. The updrift portion of each spit is shadowed by its updrift neighbour. Given that flying spit growth requires a wave climate in which the dominant wave angles are high and from the updrift direction, this shadowing is greatest at the updrift end of each spit. The shadowing decreases progressively downdrift, the spit coastline becoming more exposed to the waves approaching from the dominant direction. Gradients in net sediment flux arising from down-drift decline in wave-shadowing effects tend to produce erosion, resulting in seaward concavity in the coastline (Figure 6b,d). However, the development of concavity is limited because the increasing curvature of the coastline tends to result in accretion as the change in coastline geometry interacts with the locally-experienced wave climate. Thus, the erosion tends to be balanced by coastline flattening induced by the local wave climates that result from wave shadowing. The concavity tends to be flattened and the locus of erosion propagates downdrift to the spit flank and tip. As long as different parts of the spit translate alongshore at different rates, the shape evolves (Ashton et al., 2016). However, when the shape and the gradients in net sediment flux adjust such that each segment of the spit translates alongshore at the same rate (given by the local cross-shore erosion rate/local coastline angle), the shape becomes persistent in a (dynamic) steady state; this steady state is characterised by episodic spit loss and subsequent rearrangements of the remaining spits.

When the wave climate shifts to one still asymmetric in character, but with an increased proportion of low-angle waves (e.g. see wave rose pdfs in Figure 3 for $U = 0.55$ & $0.45$), the net sediment flux will still be directed toward the tip of the spit (cf. capes where the flux is directed away from the tip). Thus, rather than contributing directly to a reduction in aspect ratio, the positive net flux continues to be captured at the spit tip. However, because the shoreline orientation experiencing the maximum net flux is different after the change in wave climate, the spit tip propagates in a new net direction. The degree of offshore propagation is reduced (indeed, it can be directed onshore when $U < 0.5$) and the alongshore component increases. In addition, shadow zones caused by the updrift neighbouring spits shift in location and intensity as spit morphologies

change. The combined changes in flux and transport direction at each end of a spit result in flux gradients that cause the spit shape to shift gradually to a more shore-parallel orientation, in the manner described above.

Thus, for flying spits, the flux gradients scale approximately with the maximum net sediment flux divided by the total spit wavelength, rather than the small fraction of cape wavelength represented by cape tips. These two modes of adjustment –

one in which gradients in net sediment flux occur over some small fraction of the wavelength, and one in which they occur over the whole wavelength — do not apply exclusively to capes and spits, however. Once an adjusting flying spit reconnects with the down-drift coastline (Figure 3b.ii, second panel), gradients in sediment flux occurring over a small fraction of the wavelength begin to exert an influence (Nienhuis et al., 2013). Furthermore, capes formed by slightly asymmetric wave climates (Ashton and Murray, 2006a), which produce a net alongshore sediment flux, also migrate alongshore and are,

therefore, also subject to the constant alongshore-translation condition.

### 5.2 Limitations and implications

The experiments reported here involve only two types of coastline morphology and two types of wave climate change. This limited exploration motivates a wider, more systematic investigation of the responses of a broad range of morphologies to changes in wave climate. Although beyond the scope of this initial study, experiments like those depicted in Figure 2C could

be conducted for morphologies within the ($U$, $A$) phase space (Ashton and Murray, 2006a, figure 9a), with different trajectories through that space. However, even the initial results presented here show that (explicit or implicit) assumptions common in previous analyses of coastline shapes (Ashton and Murray, 2006b; Kaergaard and Fredsoe, 2013b; Moore et al., 2013; Ribas et al., 2013; van den Berg et al., 2012; Idier and Falqués, 2014), or adjustments to wave-climate change (Barkwith et al., 2014; McNamara et al., 2011; Slott et al., 2006) could be wrong. Coastline morphology should not be

assumed to be in quasi-equilibrium with the current forcing; rather it could instead represent a legacy of past forcing conditions, possibly from many centuries ago.

The results reported here provide *guidelines* for critical timescales of wave-climate change below which either capes or spits would fail to respond in a quasi-equilibrium fashion. These set limits on the mode of coastline adjustment that will occur: when wave climate changes occur over timescales shorter than the characteristic timescale for morphological adjustment, the

shoreline response will exhibit significant morphologic memory. In contrast, longer timescales for change permit a quasi-equilibrium response, in which morphological adjustment keeps pace with the change in wave climate. This scaling makes it possible to extrapolate the temporal limits on equilibrium adjustment described above, permitting distinction between quasi-equilibrium and long-term memory response of coastline features at arbitrary spatial scales.

These critical timescales depend not only on coastline morphology and the scale of coastline features (see also Hurst et al.,

2015), but also on the characteristic wave heights, shoreface depths, which influence rates of coastline change in quantifiable ways. The timescales predicted by model experiments, such as those we present here, will be altered quantitatively when different wave heights, shoreface depths, or alongshore-sediment flux relationships (or, indeed, empirical coefficient values)

are used. However, such quantitative changes will not affect the relevance of comparing timescales for morphological response and wave-climate change to understand or forecast whether coastlines will exhibit a quasi-equilibrium response or be influenced by a 'memory' induced by preceding morphological states.

These results also have implications for management of potentially fragile sandy coasts. Management policies and plans are commonly underpinned by predictions of future shoreline erosion (or accretion) rates along developed coastlines that are based on shoreline change observed over previous (usually very few) decades. Observations accumulated over such relatively short timescales may not be sufficient to discern the true direction of morphological change, since these are of roughly the same order as the characteristic timescales indicative of limits to the potential for equilibrium morphological response, as calculated above. Furthermore, change in environmental drivers (weather patterns, storm frequency, etc) may be poorly understood. Thus, wider analysis of environmental conditions and coastline response is likely needed for more informed decision making. Indeed, coastlines deemed to be under threat from climate change effects, and therefore requiring socio-economic as well as environmental management should benefit from long-term monitoring of weather and geomorphology to understand what kind of intervention might be necessary, and to help preclude costly, non-beneficial or even damaging actions.

Our analysis has purposefully not considered changes in cross-shore sediment flux resulting from erosion related to sea level rise (e.g. Moore et al., 2010; Wolinsky and Murray, 2009). However, the changes in coastline shape we have addressed can be superimposed on shoreline change associated with sea-level rise (e.g. Bruun, 1962; Walkden and Hall, 2005, 2011). Increases in the rate of sea-level rise generally cause increases in shoreline erosion, but the response to sea-level rise is approximately uniform along a sandy coastline at the scales of interest (e.g. Moore et al., 2010). In contrast, planform changes in coastline position arising from alongshore gradients in sediment flux indicate heterogeneous erosion and accretion along a shore: spatial patterns and magnitudes of accentuated shoreline erosion could be different in the future, compared to the recent past. This could result from quasi-equilibrium style adjustment to future changes in wave climate, analogous to that which Moore et al. (2013) found for cuspate capes over recent decades. However, our results illuminate the possibility that even without wave-climate change occurring in the present or near future, coastline shape could be in the midst of a long-term readjustment to changes in wave climate that occurred in the past. Given the diffusive scaling of coastline response timescales with length scales, large coastal features inherited from the early Holocene may not yet have adjusted to current wave conditions. In such a case, zones of accentuated erosion would tend to shift in location and intensity over time, without any warning from identifiable changes in forcing. This possibility motivates future work to develop metrics in terms of alongshore patterns of local wave climates (e.g. Ashton and Murray, 2006b) or sediment fluxes that could identify coastlines that are in or out of (quasi-) equilibrium with the present directional wave climate.

Finally, the results of these model experiments might also have implications for geologic and paleo-climate interpretations. Our modelling results indicate that flying spits adjusting to changes in the wave angle distribution in a wave climate can

leave behind records of the adjustment in the form of complex arrays of lagoons enclosed by beach ridges (e.g. see Figure 3b.ii, second panel, 3b.iii, first and second panels). In natural settings, the lagoons, which can potentially extend far enough landward to be preserved, would fill with fine sediment over time. However, the morphological arrangement of such lagoons and associated bounding beach ridges, preserved in the geological record, could indicate the effects of changing wave climate on a spit coastline. Such complicated coastal plain deposits can form in other ways (including reworking of a relict, potentially crenulated coastline present at the beginning of the current sea-level high stand, and/or episodic fluvial or coastal sediment delivery (Nienhuis et al., 2015)), but being aware of the possible influence of wave climate change on the morphology and structure of coastal hinterlands could inform reconstructions of coastal geographies and paleo-wave climates.

**Conclusions**

We have explored the degree to which complex sandy coastlines can adjust in a quasi-equilibrium manner to changing wave climate, and the degree to which past wave climates, as manifest in existing morphology, might influence subsequent coastline evolution during wave climate change. We conducted numerical modelling experiments in which capes and spits grown under some initial condition were subjected subsequently to a different wave climate. We find that the resulting evolution of complex coastlines (capes and spits) is influenced by previous morphology for a significant period of time - in some cases rather longer than indicative characteristic timescales of morphological change (few decades to few centuries) after wave climate change has occurred. Characteristic timescales for change show that spits respond at rates approximately an order of magnitude more slowly than capes, and these timescales depend on the spatial scale of the coastline feature (a diffusive scaling). In particular, quasi-equilibrium response cannot be assumed: such behaviour will depend on the rate of wave climate change, and how it compares with the characteristic timescale for morphological adjustment for a given coastline morphology. Significant changes in wave climate on decadal to centennial scales may result in hysteresis in the response of the coastline morphology. Thus, modern coastlines may be out of morphological equilibrium with respect to current wave conditions, reflecting instead the wave climates to which they have been subject in the past, and the resulting antecedent coastline morphology. The preservation of beach land forms in the coastal hinterlands behind modern sandy coasts may reflect this history and provide insights into past palaeo-wave climate and coastal geography.

**Author contribution**

CWT and ABM planned the experimental design and undertook the modelling and research, and prepared the manuscript, supported by ADA and MDH. The original ideas for this work, and their subsequent development, grew out of initial discussions between CWT and AKAPB; AKAPB subsequently provided further ideas, advice and support for the later

modelling. MAE supported and helped shape the research through discussion and further ideas. All have contributed to the manuscript through additional or modified text and comments.

**Acknowledgments**

CWT gratefully acknowledges support from the British Geological Survey and the Nicholas School of the Environment, Duke University, NC, for a short sabbatical at Duke University in 2015, during which this work was largely undertaken. Dr Eli Lazarus and an anonymous referee are thanked for constructive and insightful reviews. This work was funded by NERC National Capability core funding to the British Geological Survey. The Coastline Evolution Model (CEM) can be downloaded from the Community Surface Dynamics Modeling System model repository.

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

**Figure Captions**

**Figure 1**

Diverse large-scale morphologies developed on open sandy coastlines under the influence of dominantly high-angle offshore waves ($U > 0.5$). Spits develop under an asymmetric distribution of wave angles ($A \sim 0.7$); nearly symmetric capes develop where the wave angles are distributed symmetrically ($A \sim 0.5$). Representative wave angle distributions are taken from wave buoy data available for these examples. Namibia: after Bosman and Joubert (2008); Carolina capes: WIS Station 509; after

Ashton and Murray (2006b); North Norfolk, UK: data from Blakeney Overfalls Waverider III buoy, CEFAS WaveNet.

**Figure 2**

2a. CEM model scheme. The coastline is discretised into cells. Local shoreline angles, $\theta$, at each cell are calculated according to the schema outlined by Ashton & Murray (2006a). Sediment transport is determined by the effective wave

angle ($\emptyset - \theta$). Arrows indicate net flux direction under waves incoming from the left; arrow lengths qualitatively indicate the flux. Sand is not transported through cells which are in shadow for a particular wave. Barrier width is maintained by a simple overwash scheme described by Ashton and Murray (2006a).

2b. Rose diagrams of the four-bin probability density functions for wave climates used in the modelling, based on parameters $U$ and $A$.

2c. Wave climate trajectories used in the simulations defined by the $U$ and $A$ phase space, after Ashton and Murray (2006a). Wave climate asymmetry is defined by $A$. $U$ defines the proportion of offshore high angle waves ($> 45°$). **C**: cuspate capes; **R**: recurved, connected spits; **S**: flying spits; **SW**: large-scale sand waves.

**Figure 3**

Snapshots of simulated coastline morphologies evolved under changing wave climate. See text for discussion.

ai – iii: capes evolved under a symmetric wave climate.

bi – iii: spits evolved under an asymmetric wave climate.

Horizontal and vertical scales are the same for all, representing a domain 30 x 200 km. The top-most two panels show the coastlines evolved for 250 years under initial conditions of $U = 0.7$ for $A = 0.5$ (capes) and 0.7 (spits) respectively. The next

two blocks of three panels labelled with the $U$ values of the changed wave climate show, respectively, the coastline morphologies evolved 200 and 500 years *after* the wave climate is changed at 250 years, and the morphologies evolved over 1000 years under static wave climates with the same $U$, $A$ values as the changed wave climate (Note that in these examples,

the wave climate is changed gradually over 100 years.) The greater length-scales in the morphologies of the coastlines altered by the changed wave climates are clear. The persistence of cross-shore amplitude for a spit coastline evolving under $U$ of 0.45 is particularly evident. Model scales are the same in all panels. PDFs of the wave climates are included for comparison.

**Figure 4**

Time-series of the evolution of aspect ratio (amplitude/wavelength), wavelength and amplitude for capes (ai – iii) and spits (bi – iii) under 100 years of *gradual* change in wave climate, from 250 years. In all diagrams: Dark blue line: results for the coastline evolved under a static wave climate with $U = 0.55$, for both spits and capes. Cyan line: progressive change in $U$

from 0.7 to 0.45 over 100 years. Magenta line: progressive change in $U$ from 0.7 to 0.55 over 100 years. e-folding times are shown graphically for 900 year models (see Figure 5). These relate *specifically* to change from $U = 0.7$ to $U = 0.45$, based on change in aspect ratio and are shown to provide additional context for timescales of gradual change. See text and Supplementary Information. **Note**: Data are smoothed by a 7-point moving average to aid clarity. The indication of change in morphology slightly before 250 years is an artefact of this smoothing.

**Figure 5**

Time-series of the evolution of aspect ratio (amplitude/wavelength), wavelength and amplitude for capes (ai – iii) and spits (bi – iii) under *instantaneous* change in wave climate at 250 years (900 year models). In all diagrams: Dark blue line: the results for the coastline evolved under a static wave climate with $U = 0.55$, for both spits and capes. Cyan line: $U$ changing

from 0.7 to 0.45. Magenta line: $U$ changing from 0.7 to 0.55. e-folding times are shown graphically for the 900 year models. These relate *specifically* to change from $U = 0.7$ to $U = 0.45$, based on change in aspect ratio. See text and Supplementary Information. **Note**: Data are smoothed by a 7-point moving average to aid clarity; cf. Figure 4.

**Figure 6**

Detailed mapping of potential net flux and stability index across example coastline features, for different wave climates impacting on static (un-evolving) coastlines. a, b: net flux data for capes (a.i) and spits (b.i); gradients in the *difference* between the net fluxes for capes (a.ii) and spits (b.ii) c,d: Stability index data for capes (c) and spits (d). magenta line: $U = 0.7$; green line: $U = 0.45$; yellow dashed line: difference in potential net flux. Rose diagrams of the wave climate pdfs are included for comparison.


**Figure S1**

The metrics for capes used in the calculations of the diffusive scaling relationship between wavelength and time. The figure accompanies Table S1.

**Figure S2**

5    The metrics for spits used in the calculations of the diffusive scaling relationship between wavelength and time. The figure accompanies Table S2.

Figure 1

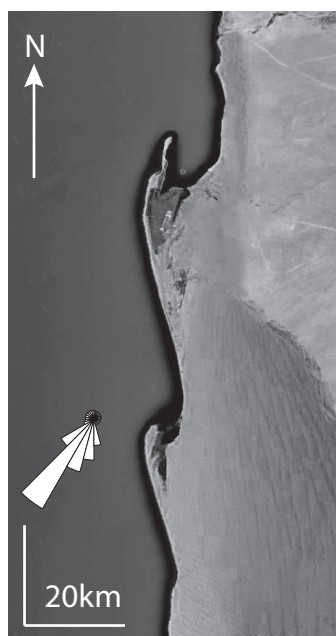

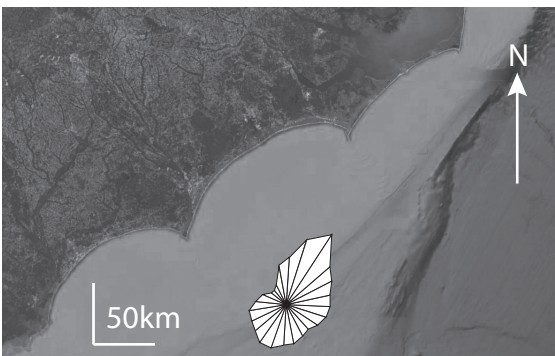

Capes, North Carolina coast, USA

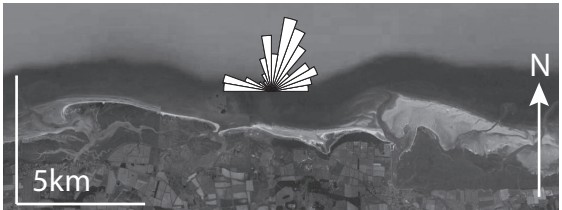

Flying spits, Namibia      Low-angle connected spits, Norfolk, UK

Figure 2

## (a)

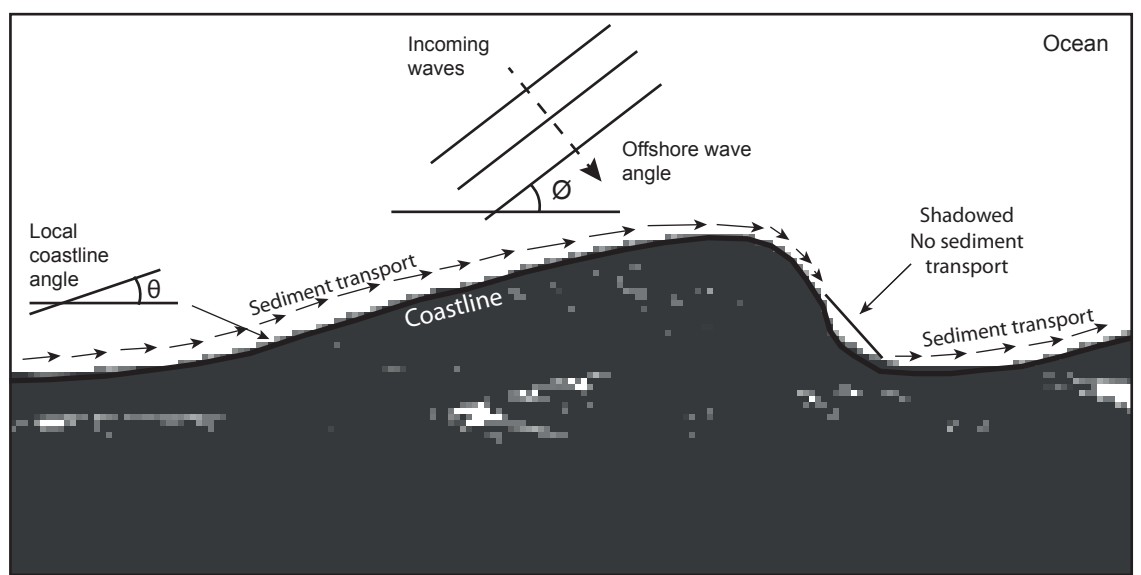

## (b)

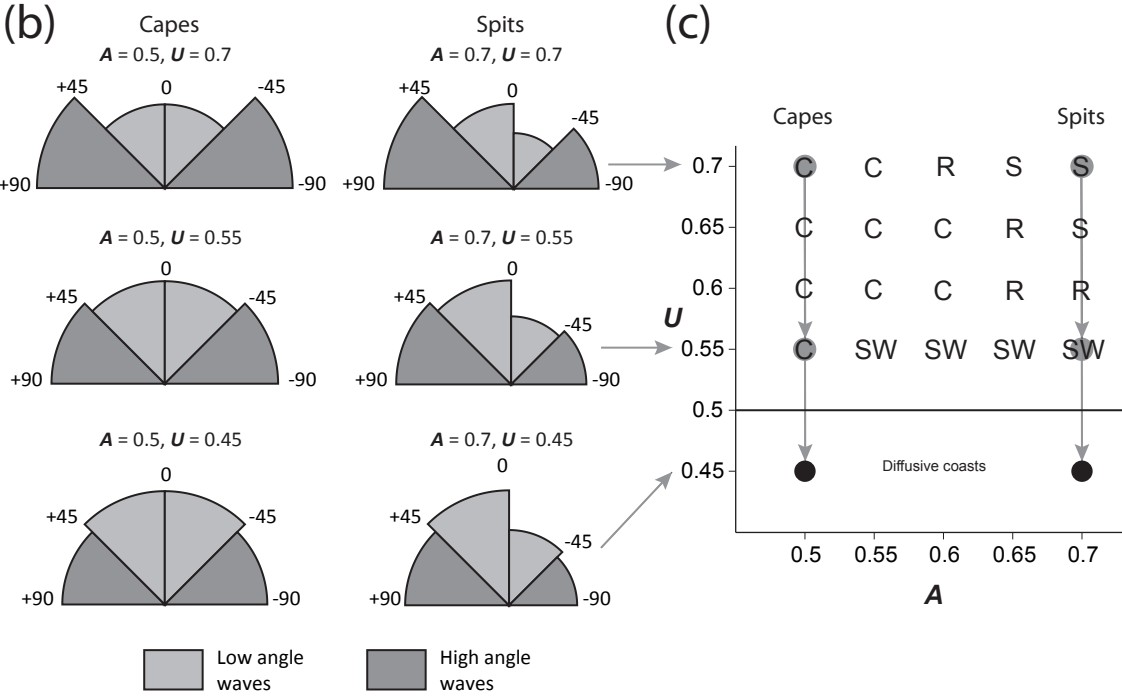

a: CEM model scheme
b: Wave climate probability density functions
c: Wave climate phase space

Figure 3

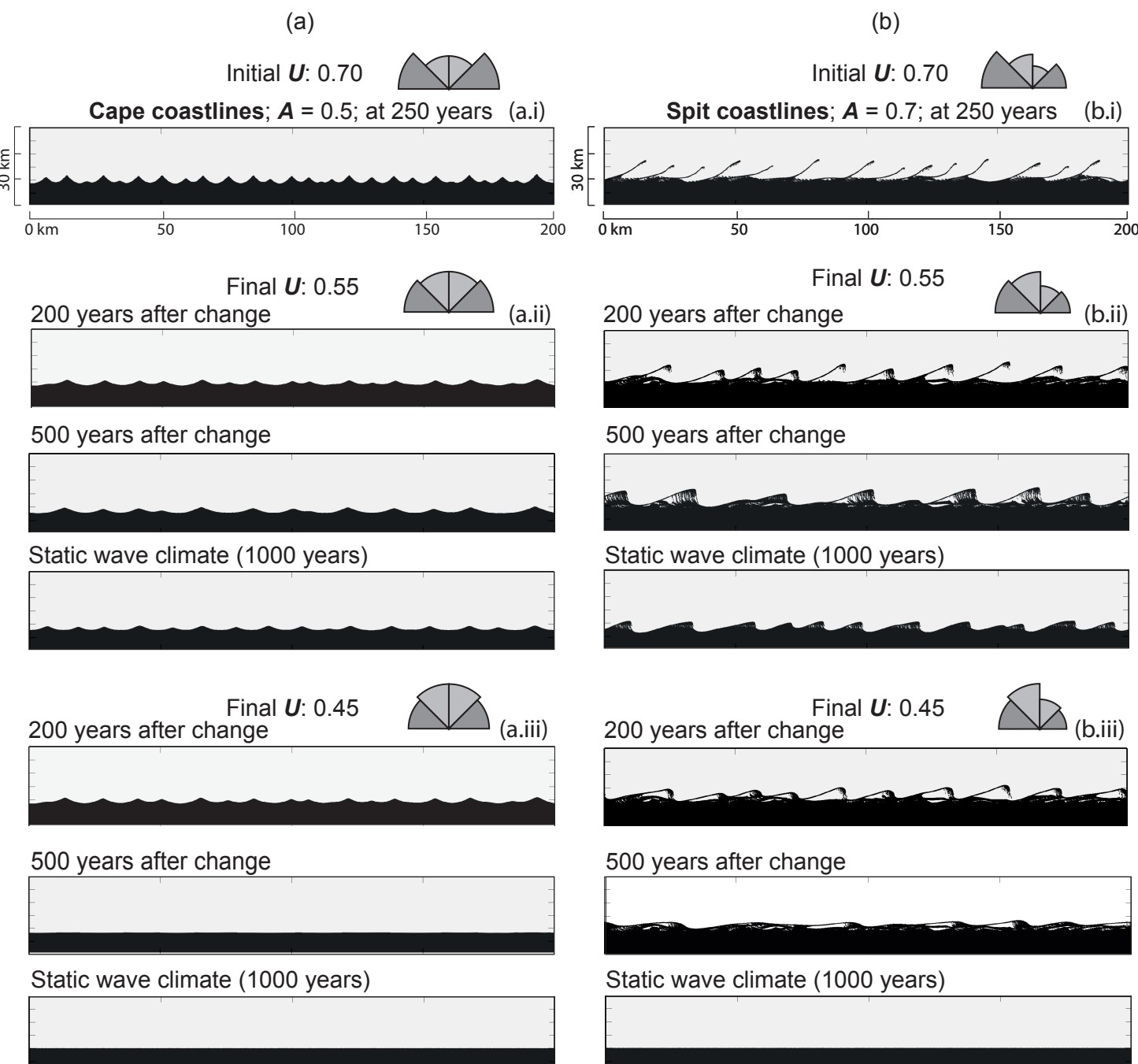

Figure 4

## Capes; *A* = 0.5

(a.i)

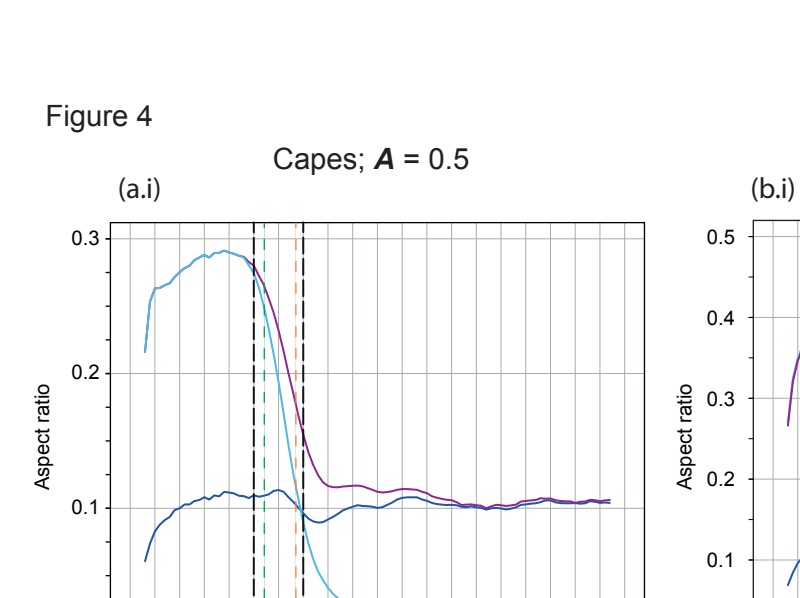

(a.ii)

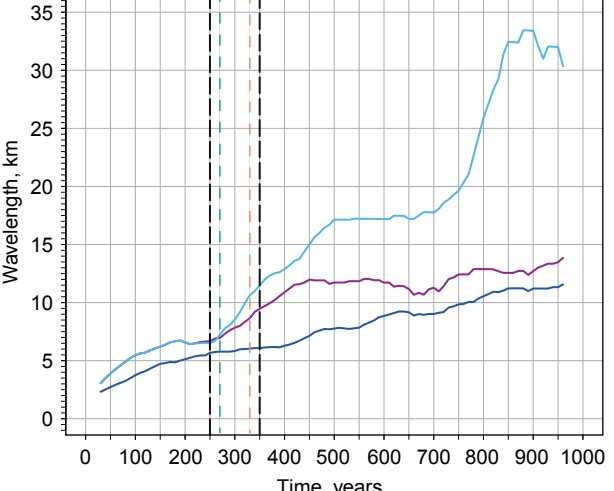

(a.iii)

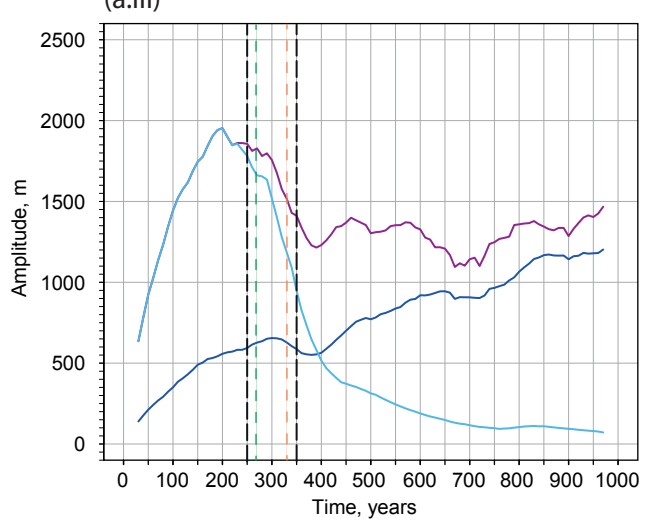

## Spits; *A* = 0.7

(b.i)

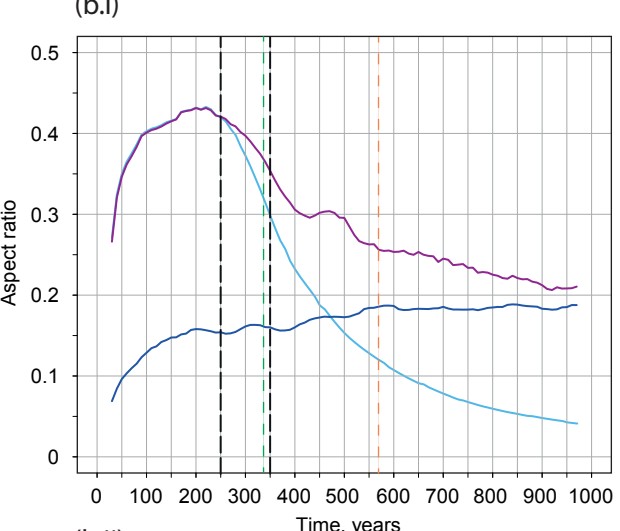

(b.ii)

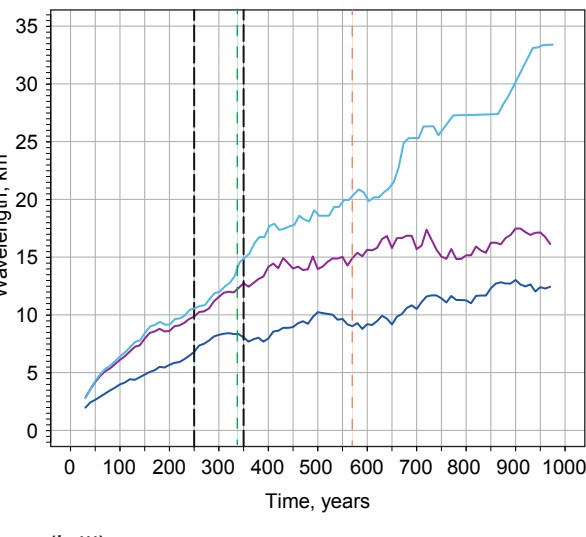

(b.iii)

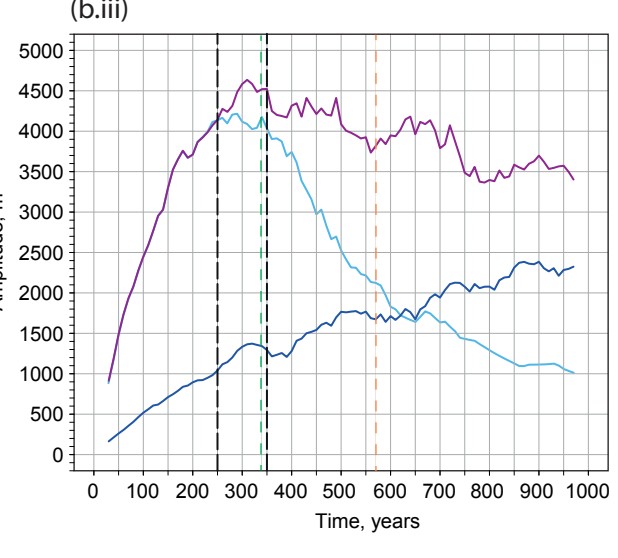

——— Static *U* = 0.55    ——— *U* = 0.7 to 0.55    ——— *U* = 0.7 to 0.45

– – – – – – Time interval over which wave climate is changed

e-folding times for changes in aspect ratio for *U* changing *instantaneously* from 0.7 to 0.45 (see Figure 5)

– – – – – 1st e-folding time for 900 year models: capes: 20 years; spits: 90 years

– – – – – 2nd e-folding time for 900 year models: capes: 80 years; spits: 320 years

Figure 5

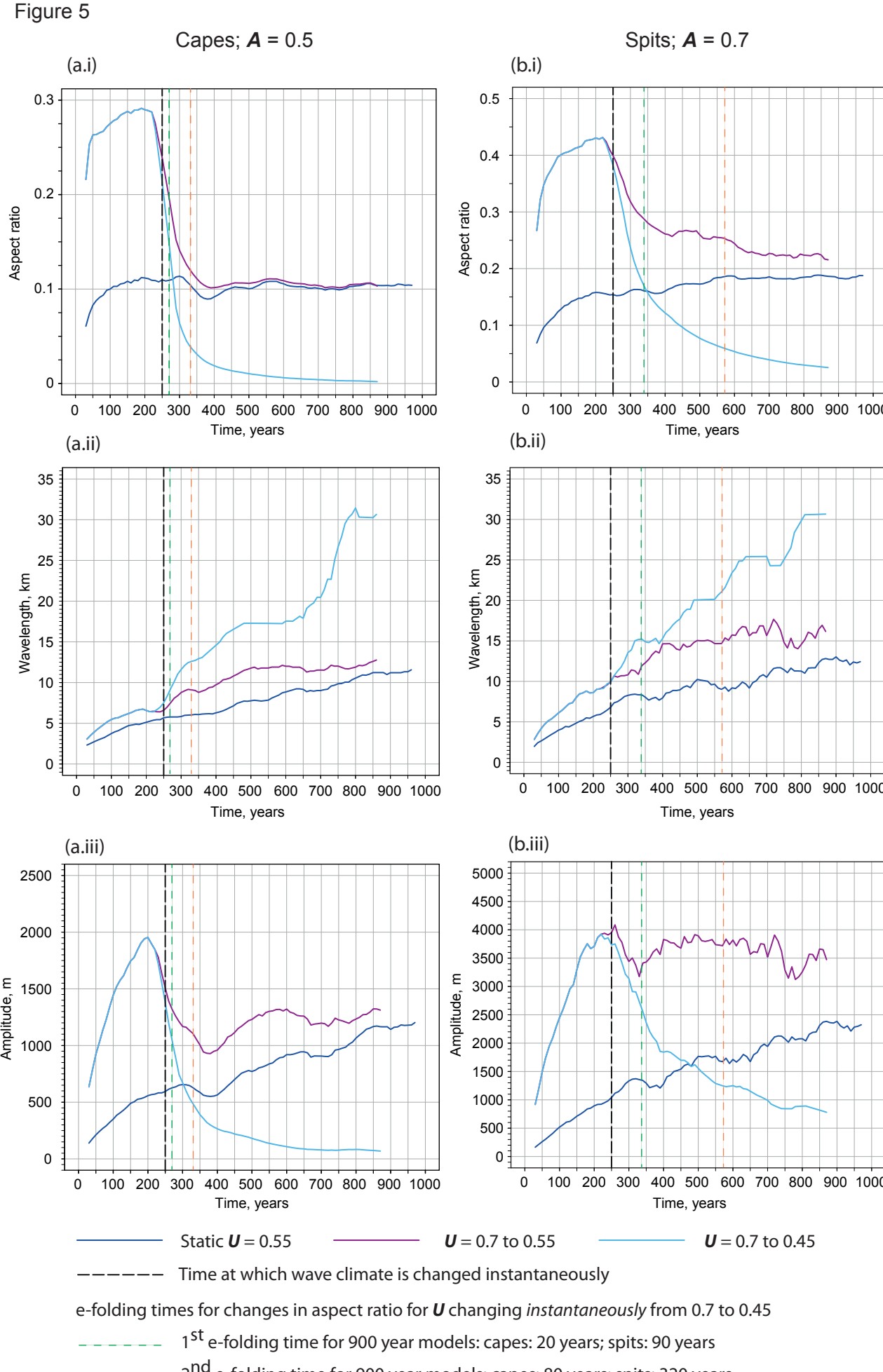

Capes; *A* = 0.5

Spits; *A* = 0.7

(a.i)

(b.i)

(a.ii)

(b.ii)

(a.iii)

(b.iii)

Static *U* = 0.55          *U* = 0.7 to 0.55          *U* = 0.7 to 0.45

— — — — —  Time at which wave climate is changed instantaneously

e-folding times for changes in aspect ratio for *U* changing *instantaneously* from 0.7 to 0.45

— — — — —  1st e-folding time for 900 year models: capes: 20 years; spits: 90 years

— — — — —  2nd e-folding time for 900 year models: capes: 80 years; spits: 320 years

Figure 6

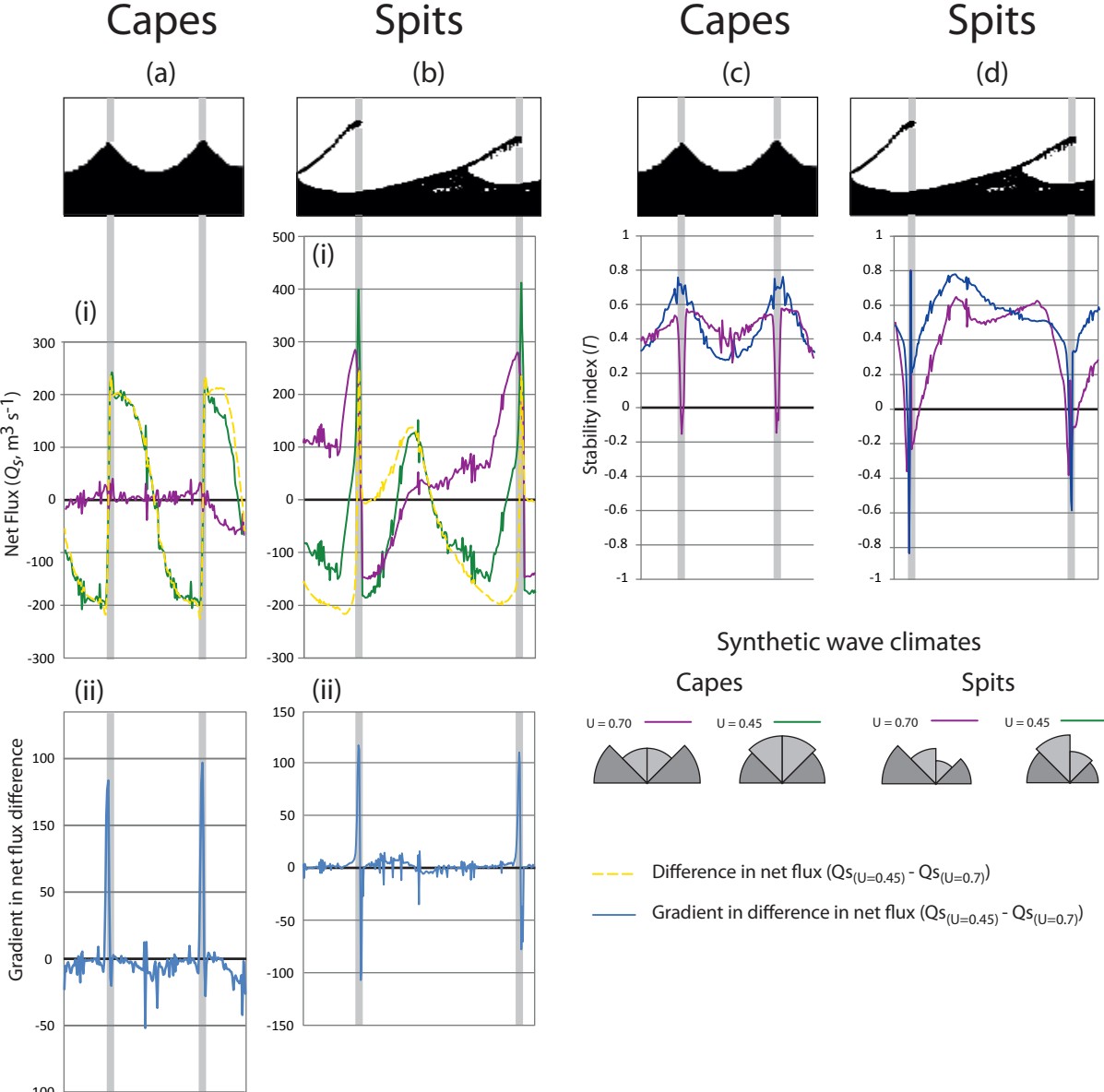