# Peer review of "Complex coastlines responding to climate change: do shoreline shapes reflect present forcing or 'remember' the distant past?"

_Earth Surface Dynamics, 2016_

## Referee Comment (RC1) · E. D. Lazarus (Referee) · 14 Aug 2016

REVIEW – Complex coastline response: Thomas et al. (esurf-2016-35)

In this paper, the authors use the established Coastline Evolution Model (CEM) to explore morphological legacies of two different kinds of wave-dominated, "complex" coastal landforms – cuspate capes and flying spits – under a shift in forcing conditions from constructive to destructive. In the CEM, capes and spits are emergent, self-organized features (hence "complex") that develop in response to a high-angle wave climate (a predominance of deep-water waves with an angle of incidence $>\sim45°$ and $<90°$ relative to the shoreline). Where a high-angle wave climate is thus constructive,

a low-angle wave climate has the opposite effect, driving diffusive smoothing of the coastline.

The authors present a numerical modelling experiment in which two complex coastlines, one characterized by capes and the other by flying spits, that form under high-angle wave climates are then confronted with low-angle wave climates. How long does it take for the existing (antecedent) morphology to diffuse away (if it ever does)? That response time is converted into a "characteristic time scale" of coastal morphological adjustment (for a given morphology). If the characteristic time scale for morphological adjustment is shorter than the time scale over which the forcing conditions change, the coastline maintains a state of quasi-equilibrium. (Here, that state is defined in terms of observed feature aspect ratios relative to those expected from the forcing applied.) Oppositely, if the characteristic time scale for morphological adjustment is longer than the time scale over which the forcing conditions change, then antecedent conditions impart a legacy effect on how the coastline evolves, resulting in observed features with shapes and spatial scales significantly different from those expected from the forcing.

This paper represents an important examination and extension of CEM behaviour, and I fully support its publication in ESurf. However, the summary above was challenging to assemble – and that is the primary aspect of this manuscript I encourage the authors to address. I didn't find this draft particularly easy to read. I think the "why" of the paper gets crowded out by the technical "how", obscuring the through-line and rationale underpinning much of the "what".

Part of this problem may originate in the Introduction. The second paragraph mentions "different wave climates", "shifts wave climate", and "slight differences in wave climate" without being more specific about what those shifts (directionality? angularity?) are or might be relative to present conditions. The reason the specificity of this framing matters is that the modelling experiment itself is very specific. The authors begin with complex coastlines created by high-angle wave climates and then only dial that forcing back (from high-angle to lower/low-angle wave climates) – they never turn the forcing

up, or oscillate between high- and low-angle wave climates. The carefully defined focus of the experiment is good. But the Introduction suggests, with its broad scope, that these two different coastline shapes perhaps will be battered with a variety of wave-climate shifts, rather than an effectively single shift (slow or fast) from high-angle to low. The end of the Introduction states that the "wave angle distribution was changed..." [P2, L25], but the authors don't specify the character of that change (anti-diffusive to diffusive) until [P4, L5–10]. Furthermore, after this initial mention of "the transition from anti-diffusive to diffusive wave climates" at [P4, L9], explanation of the forcing "trajectory" from anti-diffusive to diffusive [P6, L17] doesn't come for another two pages, deep in the experimental design (Section 3.1).

A clear paragraph at the end of the Introduction that frames the experiment more specifically (not just that the wave climate gets adjusted, but gets adjusted in this way...) would propagating through the sections that follow. Such a paragraph would also give the authors an opportunity to provide some up-front rationale for their experimental design before they explain the design in all its nuts and bolts. (In the paper's present sequence, the time frames described at [P5, L20–30] come across as arbitrary choices rather than deliberate.) Why only change the wave climate from anti-diffusive to diffusive? That reasoning may be straightforward to someone familiar with the model (and therefore skipped over), but I think playing out that thought experiment here is both worthwhile and necessary, given the paper's orientation with regard to management projections and shoreline-change forecasting.

I think it might also help for the authors to make clear early in the manuscript what "dynamic equilibrium" (or quasi-equilibrium) means in this system – that for a given combination of parameters U and A, the model eventually delivers a coastline configuration characterized by a certain pattern and morphometrics, with behaviour quantifiable at every point alongshore via gamma. "Dynamic equilibrium" appears often in the first half of the manuscript (as the initial coastline configurations get spun-up), but readers may not have a physical sense of what that equilibrium is or why it arises. Indeed, a section

on dynamic equilibrium – perhaps after Section 2.2, having introduced gamma as a metric – may offer a home for the "mechanism" sections 5.1.1 & 5.1.2, which I found out of place in the Discussion. The first two-thirds of both 5.1.1 & 5.1.2 are dedicated to what the mechanisms are in an anti-diffusive regime, anyway. Consolidating these various sections would allow the remaining "after the shift" elements of 5.1.1 & 5.1.2 to move into the Results (because that's really what they are), somewhere around 4.1.1 & 4.1.2.

One way to paraphrase all of my suggestions above is to look for ways to group like with like; many related pieces are compartmentalized into disparate subsections. With focused revision in the first two thirds of the paper, the closing sections will fit more comfortably with the Results. Finally, I encourage the authors to look for ways to explain things more simply wherever possible. The paper is at its best in those moments (the Conclusion, for example).

Minor comments:

[P2, L10] I think this paragraph would benefit from a different topic sentence, and the "analogous study" remarks should move down in the paragraph. (Indeed, these two paragraphs could merge, with rewriting.)

[P2, L24] Coming after a reference to Fig. 1, the line "these coastline morphologies using the appropriate wave-angle distribution" suggests (or leaves open the possibility) that the wave climates are tuned to the examples in Fig. 1. The modelled coastlines are generic – deleting this mention is probably fine, given the space in subsequent sections dedicated to explaining the modelling conditions. Note that the discussion of "mapping gamma" at P4, L5 is likewise ambiguous with regard to Fig. 1 – the gamma explanation needs clarifying.

[P5, L10] I did not understand Section 2.3.2 – neither the explanation offered, nor its relationship to the Results.

[Fig. 3] I don't understand what the authors mean by the "static wave climate" condition. The only mentions I found appeared in the figure captions, and given the various scenarios being compared I wasn't sure what was actually "static" – an unchanging PDF? a single angularity and direction? (In either case, what is the static condition?)

[P7] Section 4.1.3 looks to me like it should lead this subsection (become 4.1.1).

[general] Look for opportunities for shorter sentences, especially given the paper's technicality? (Swap semi-colons for periods?)

[general] The authors may disagree, but I would also urge them to specify "characteristic time scales of morphological change" (or similar) rather than simply "characteristic time scales" because, here, wave climate also changes and could conceivably have some characteristic time scale of its own. I know the only characteristic time scale of interest in this work is for morphology, but phrases like "characteristic time scales for change" [P12, L23] still read as ambiguous, supported as they are by the context.

[Fig. 3] Fig. 3b.iii, second panel of the triptych – is that a "capes" run or a "spits" run? Those look to me like the result of a symmetrical wave climate. . .

[Figs. 4 & 5] Is there a way to visually incorporate the "characteristic time scales of morphological adjustment" (both the first and second e-folding times) into these panels to lend the reader that background frame of reference?

[Fig. 6] Sub-panel at bottom right looks like it might be out of justification with other elements? Unfinished relative to the others?

I look forward to seeing the final version of this manuscript in print.

Best of luck –

Eli Lazarus University of Southampton Southampton, UK

---

## Referee Comment (RC2) · Anonymous Referee #2 · 29 Aug 2016

**Review report on paper esurf-2016-35, 'Complex coastlines responding to climate change: do shoreline shapes reflect present forcing or 'remember' the distant past?, by Thomas, Murray, Ashton, Hurst, Barkwith and Ellis.**

This paper examines the morphodynamics of sandy coasts driven by wave-induced alongshore sediment transport. It is well known that for large wave incidence angles with respect to shore-normal, the coupling between wave field, morphology and sediment transport may become unstable and give rise to complex shorelines featuring sand waves, cuspate capes and spits. When trying to test this assumption against observations, all the existing literature considers the present wave climate as the forcing agent. However, the climate changes and that assumption implicitly assumes either no change or very slow change with respect to the characteristic time scale of the features.  This paper explores to which extend this assumption is valid by means of model experiments for two different features: cuspate capes and flying spits. It is found that the characteristic time of spit dynamics is long enough for the quasi-equilibrium with present climate assumption to be not valid. This implies that present coastal morphologies may respond not only to the present conditions but also to past wave and morphological conditions, possessing a sort of memory. This is relevant for coastal management, for interpretation of geological records and for paleo-climate insight. It also makes more difficult the comparison between models and observations regarding coastline features.

I find the results innovative, relevant and well-funded. The manuscript is generally clear and well-written. I therefore recommend publication after addressing some comments I list here below.

1) In the abstract it is claimed that the characteristic scales vary with the square of the aspect ratios. Where is it proven in the paper? I can't find it. Please check or remove.
2) There is some confusion regarding the duration of the model runs. According to the figure caption, "after 450 yr" is wrong in Figure 3. It should be "after 200 yr". I would change "static wave climate"$\rightarrow$ "static wave climate during 1000 yr", for the sake of clarity. I presume the 200 yr or the 750 yr include the spin up 100 yr, isn't it? Page 6, line 23: "the new wave climate is held constant for a further 600 yr". Shouldn't be for 650 yr? =1000-(250+100)?  Please clarify!
3) Page 8, line 30: "the resulting strong gradients in these fluxes are directed towards the inter-cape bays, causing cape tips to erode and bays to prograde". This sentence is formally incorrect, is the sediment flux that is directed towards the bays, not the gradient. The gradients are directed with the x-axis at the tips and against it at the bays. But there is no need to say this; the key point is that there is divergence of flux at the tips and convergence at the bays.
4) For U=0.45, spits are also smoothed out but more slowly than capes. It is said "over a time scale of many centuries". According to the modeling runs, is it possible to be

more specific? Could the authors run the model for longer time than 1000 yr? Or perhaps this could be approximately inferred from the runs already made?

5) Eq. 1. dy/dt → ∂y/∂t , dQ/dt → ∂Q/∂t . Also, it is said that "y" is the cross-shore position. This is too vague. You should specify that y(x,t) is the cross-shore position of the shoreline.

6) Line 22: the net diffusivity is the sum of the individual diffusivities. This is only true due to the linearization of the dynamic equations with respect to shoreline displacement. A recall or warning would be appropriate.

7) Eq. 3. " … individual diffusivities are calculated … using:". This might be misleading for the readers. The diffusivity is not computed "using". In this context, this is THE expression of the diffusivity obtained by Ashton and Murray 2006a. Although this reference is already cited earlier it should be emphasized that this equation comes out from this paper. Otherwise, the readers could erroneously think that this formula is straightforward. So I would rephrase a bit indicating that eq. 3 is the shoreline diffusivity obtained by Ashton and Murray 2006a.  Also, it is a bit strange talking about diffusivity and not writing down the corresponding diffusion equation. Perhaps it is better to include the equation.

8) How the time scales would change with wave height and shoreface depth? A guess? In relation with this, it is said in the conclusions that the adjustment time scales depend on the spatial scale of the feature in a diffusional scaling. However it also depends on the wave energy incident on the coast (and on the depth of the shoreface) and this should be acknowledged here for the sake of clarity. Otherwise, readers (sometimes reading only the conclusions) could be misled.

9) Eq. 2. $H_0$ is significant or rms? Please specify.

10) I find a bit strange the expression in line 13, page 4, "the principle underlying…". Which principle? Sediment conservation? One-line approximation? Perhaps something like "principles" is more appropriate.

11) Page 7, line11: "small, relatively low-amplitude reconnected spits" shouldn't be sand waves (according to diagram C in Figure 2)?

12) Page 8, line 22: "potential net flux" → diffusivity?

13) Page 9, line 1: "The fluxes are proportional to the maximum net flux divided by the alongshore length scale".  I guess the authors refer to the gradients.

14) Is the difference in sediment fluxes needed to be plot in Figure 6a,b? I think the necessary information is in the other two lines. The third one complicates unnecessarily the panels.

15) Page 10, lines 4-5. Looking at Figure 6b I also see strong gradients in sediment flux for spits. I agree in that they are not as strong as for capes, but maybe the wording in the paper is a bit exaggerated regarding this difference.

16) Figure 6b. Why the span for Q is larger than that in Figure 6a? Any line exceeds the upper value of 300. Why using 500 as upper limit?

17) Page 10, lines 21-22. "Coastline morphology should not be assumed … from many centuries ago". Not always, it depends on the time scales, in some cases the morphology can be in quasi-equilibrium with current wave conditions. Please, account for this here (it is already said elsewhere, but I miss it here).

Typos:

- Page 1, line 15: "cuspate cusps" → cuspate capes.
- Page 7, line 11: "with with" → with
- Page 10, line 3. "chore"→ shore

---

## Author Response (AR1)

**Complex coastlines responding to climate change: do shoreline shapes reflect present forcing or 'remember' the distant past?**

C. W. Thomas, A. B. Murray, A. D. Ashton, M. D. Hurst, A. K. A. P. Barkwith, and M. A. Ellis

**Detailed response to reviews by Dr Eli Lazarus and the anonymous reviewer**

1. *Response to review by Dr Eli Lazarus*

We thank Eli for an insightful, thoughtful and helpful review.

In his opening remarks following his summary, Eli feels that: a) the 'why' of the paper gets lost amidst the 'how', b) highlights a lack of clarity on the reasons for our particular approach and c) asks for more clarity with regard to what we mean by wave climate change.

We have endeavoured to modify the introduction to address the specific issues he raises and hope that our additional text clarifies our intentions. We hope that this clarifies our purpose and helps make the manuscript easier to read overall, following the introduction.

We have considered closely Eli's comments on structure and grouping of certain parts of the text. However, we feel that the way we have organised the paper makes most sense, having gone through various iterations in structure in drafts prior to submission. In addition, given that the anonymous reviewer regarded the manuscript as generally clear and well-written we are loathe to make significant changes. We feel that the structure is also consistent with other publications on similar coastal modelling, and with the explanation of modelling, mechanisms and interpretations. However, we have taken on board many of Eli's comments and made revisions to the text in several places to aid clarity. We hope that the revised introduction may help in answering Eli's misgivings in this regard.

With regard to specific comments:

P2, L10: We have modified the paragraph beginning 'In this paper…' to be more explicit about the questions we are addressing in the paper. Hopefully this gives the previous paragraph more context.

P2, L24 and P4, L5: It is not clear where the confusion lies here as the explanations seem appropriate. The paragraph on Gamma doesn't specifically refer to Figure 1 here, but such an approach was used initially by Ashton and Murray in their 2006b paper, so we have referenced this here, by comparison.

P5, L10: We have modified and expanded the text hopefully to make this section clearer and more connected. In essence, we are explaining how we garnered the net flux and diffusivity data from the CEM, in largely practical terms.

Fig 3: A static wave climate is one which doesn't change. This seems pretty self-explanatory to us. Static in this context means that $U$ and $A$ are unchanged through the run of the model. Of course, H and T are also fixed. Hence, this represents long-term static conditions.

Note that we have also corrected the second panel of the triptych 3b.iii – thank you for pointing out that the incorrect coastline had been mistakenly included.

P7 We feel that we can't really talk about length-scales before talking about the features to which we are referring. With regard to semi-colons, these are a very useful ways of adding sub-clauses to sentences without breaking the general theme.

General comment: '…characteristic timescales…': we have amplified this by specifying 'morphological change' in various places as suggested by Dr Lazarus.

Figures 4 & 5: e-folding times for the changes in aspect ratio for capes and spits subject to *instantaneous* change in wave climate from $U = 0.7$ to $0.45$ have been added graphically, as suggested.

Figure 6: This has been revised extensively, so the justification issue should have been resolved…

2. *Reply to the anonymous review*

We are very grateful to the anonymous reviewer for the positive, helpful and insightful review.

We have noted and corrected the typographical errors.

We have addressed the specific points raised by the reviewer below. With regard to the responses to certain comments, the text has been amended in several places, so it is difficult to paste in revised text. So that the reviewer can see the changes we have made, I have appended a revised version of the manuscript.

1. Characteristic timescales varying with the square of the aspect ratio

    The proof of the scaling relationship is referred to in the third paragraph of section 4.2. The results that show this relationship are given in Table S1 in the Supplementary information.

2. Confusion over the run lengths of the models in the caption to Figure 3.
    We have reviewed the caption to Figure 3 and have made some amendments to this and the text (Sections 3.3 and 4.1) which hopefully clear up any ambiguities.
    The spin up time is stated as 250 years (not 100 years).
    The second sentence in the caption states that the figures in the second panels (a.ii, b.ii) show the coastlines 200 years and 750 years *after* the wave climate was changed

at 250 years. 750 years should be 500 years. We have amended Figure 3 accordingly.

3. Referring to gradients in fluxes: If we understand the reviewer's comment correctly, it *is* the gradients in the fluxes that are important here, and their direction, as these dictate where erosion and deposition take place – the divergences and convergences in flux are the gradients. If the sediment flux was uniform over some distance (i.e. no gradient), there would be no erosion or deposition over this distance, since there would be no local loss or gain of sediment. The gradients indicate how the flux changes alongshore and, therefore, how and where the coastal morphology will change. However, we have amended the text slightly hopefully to amplify/clarify this point.

4. '…timescale of many centuries…': We could have run models for longer, but in this paper, we wished to emphasise the marked difference in the timescales of morphological response of capes and spits. In addition, we ran the models over a sufficient time to help us explore characteristic timescales for change. Given the other variables that affect timescales (wave energy, shoreface depth, etc), to which the reviewer refers elsewhere, we did not feel it necessary to be more specific about the ultimate timescales over which a spit might be smoothed compared to a cape.

5. Equation 1: partial differential symbols: The reviewer is correct that these should be partial differentials, since $y$ is f($x,t$). In quoting the equation from Ashton & Murray (2006a) we had simply followed their style. We have amended Equation 1 to show partial differential symbols.
We have aimed to clarify $y$ as the cross-shore position at long-shore position $x$ at time $t$.

6. Line 22: net diffusivities: the text has been amended to clarify this.

7. Equation 3: we have amended the text to clarify that this equation was derived by Ashton & Murray in their 2006 a&b papers.

8. Timescale variations with wave height and shoreface depth: We feel that paragraph 3 of Section 5.2 acknowledges this point sufficiently.

9. *Significant* wave height is now explicitly stated.

10. Wording amended to clarify.

11. 'reconnected spits' changed to 'sand-waves' and reference made to the U,A phase space in Figure 2.c.

12. Page 8, line 22: corrected – this should have been net flux, $Q_s$

13. 'The fluxes are proportional…': we have endeavoured to clarify this point in the text.

14. We have considered Figure 6 and reviewed at some length. We have revised the way we have plotted the data in this figure and added an extra panel which shows the *gradient* of the difference in the net fluxes generated by the two different wave climates, in order to emphasise the contrast in fluxes. This is augmented by additional explanatory text that hopefully clarifies the differences in the mechanistic responses of capes and spits to the changed wave climates.

15. See the comment above. It is the way the gradients in the fluxes vary over the critical parts of the spits and capes that is key to the difference in the response of the two morphologies, hence the additional panel in Figure 6. The caption has been amended accordingly.

16. The range in values in Figure 6b is necessary as the differences in the Q values do exceed 300.

17. It is not clear that any change is necessary here. Yes, a morphology could be in quasi/dynamic equilibrium with the current wave climate; this possibility is discussed earlier in the paper, where we note that the critical thing is the rate of rate of wave climate change compared to the 'ability' of a coastline morphology to respond. However, we simply re-state that this equilibrium should not be automatically assumed. This can now be tested to a first order, using the approaches we have outlined.

**Complex coastlines responding to climate change: do shoreline shapes reflect present forcing or 'remember' the distant past?**

Christopher W. Thomas[1], A. Brad Murray[2], Andrew D. Ashton[3], Martin D. Hurst[4], Andrew K. A. P. Barkwith[4], Michael A. Ellis[4]

[1]British Geological Survey, Lyell Centre, Edinburgh, EH14 4AP, Scotland, UK.
[2]Division of Earth and Ocean Sciences, Nicholas School of the Environment and Earth Sciences and Center for Nonlinear and Complex Systems, Duke University, Durham, North Carolina, 27708, USA.
[3]Department of Geology and Geophysics, Woods Hole Oceanographic Institution, Woods Hole, Massachusetts, 02543 USA.
[4]British Geological Survey, Nicker Hill, Keyworth, Nottingham, NG12 5GG, England, UK.

Correspondence to: Christopher W. Thomas (cwt@bgs.ac.uk)

**Abstract.**

A range of planform morphologies emerge along sandy coastlines as a function of offshore wave climate. It has been implicitly assumed that the morphological response time is rapid compared to the time scales of wave-climate change, meaning that coastal morphologies simply reflect the extant wave climate. This assumption has been explored by focussing on the response of two distinctive morphological coastlines - flying spits and cuspate capes – to changing wave climates, using a coastline evolution model. Results indicate that antecedent conditions are important in determining the evolution of morphologies, and that sandy coastlines can demonstrate hysteresis behaviour. In particular, antecedent morphology is particularly important in the evolution of flying spits, with characteristic timescales of morphological adjustment on the order of centuries for large spits. Characteristic timescales vary with the square of aspect ratios of capes and spits; for spits, these timescales are an order of magnitude longer than for capes (centuries vs. decades). When wave climates change more slowly than the relevant characteristic timescales, coastlines are able to adjust in a quasi-equilibrium manner. Our results have important implications for the management of sandy coastlines where decisions may be implicitly and incorrectly based on the assumption that present-day coastlines are in equilibrium with current conditions.

**Comment [cwt1]:** Anon comments that they can't find the proof of this, but it is referred to in the text and the data, etc are included in the supplementary information, so we think this is OK.

[revised manuscript text omitted]

**Comment [cwt8]:** I've revised the text to try and clarify following Eli's comment that he didn't understand this section.

**3 Experiments with changing wave climate**

**3.1 Experimental design**

**3.1.1 Instantaneous and gradual wave climate change**

We set up experiments by growing either flying spits or cuspate capes ('capes' from here-on) from an initially straight coast (with small white noise perturbations) over an initial fixed period of time. In most experiments, this initial period was 250 model years. This timeframe allows these morphologies to attain length-scales commensurate with those observed along real coastlines. We then subjected these model coastlines either to a gradual change in wave climate, or an instantaneous change. In experiments with gradual wave climate change, the initial wave climate was evolved linearly towards the new state over an arbitrary period of 100 years. These experiments were used to explore the influence of pre-existing morphology on the nature and rate of response of a coastline to changing wave climate.

In experiments involving instantaneous change, the wave climate is transformed to the target state immediately following the period of initial growth; in these experiments, we also used initial periods of 50 and 125 years to provide additional data that we could use to determine characteristic timescales for change. This allowed us to explore the possibility of scaling relationships between time and the rate of change of length scales, and the degree to which a quasi-equilibrium response in morphology is possible for given rates of wave climate change.

In both cases, coastline morphology is in dynamic equilibrium with the initial wave climate just before the wave-climate transition begins. As the wave climate changes, the coastline progressively approaches a new morphological state, settling into dynamic equilibrium with the final wave climate. We characterized coastline morphology using the aspect ratio (cross-shore extent/alongshore wavelength) of coastline features: previous work has shown that for equilibrium coastline shapes, aspect ratio varies with wave climate (Ashton and Murray, 2006a). As a coastline continues to evolve under a constant wave climate, the scale of

coastline features grows (Ashton and Murray, 2006a; Ashton et al., 2001). However, aspect ratio remains constant even as length-scales increase (Ashton and Murray, 2006a). Hence, aspect ratio is an appropriate measure reflecting the degree of dynamic equilibration with respect to a particular wave climate.

**3.1.2 Experimental wave climates**

Model simulations were driven by wave approach angles drawn from a probability density function (pdf) defined by two parameters (Ashton and Murray, 2006a): $U$, the fraction of waves approaching from angles $> 45°$ (representing the fraction of wave influences on alongshore transport from these angles) (Figure 2b) and $A$, the fraction of waves approaching from the left (CEM convention); the wave climate is asymmetric when $A > 0.5$. When $U > 0.5$, the model coastline experiences instability and perturbations will grow. The pdfs used in our modelling are shown in Figure 2b.

Ashton and Murray (Ashton and Murray, 2006a, figure 9) mapped different coastline shapes that emerge for different values of $U$ and $A$. We have explored two trajectories across the ($U$, $A$) parameter space in our experiments, one for capes and one for spits (Figure 2c). In both cases, the trajectory is towards a diffusive wave climate, under which perturbations are smoothed.

**3.1.3 Experimental conditions**

Model capes are generated over 250 years with $U$ at 0.7, and $A$ at 0.5 (Figure 3a.i); flying spits are generated with $U$ at 0.7 and with $A$ set to 0.7 (Figure 3b.i). Subsequently, $U$ is changed from 0.7 to 0.55 (moderately anti-diffusive), or to 0.45 (diffusive) while holding $A$ constant (0.5 for capes; 0.7 for spits); $U$ is changed either instantaneously, or gradually over 100 years. The new wave climate is then held constant for a further 650 years. Total model run times are 900 years for models in which the wave climate is changed instantaneously, and

1000 years for models in which the wave climate is changed gradually over 100 years. In addition, to investigate how response times vary with the spatial scale of the features, in separate experiments we generated capes and spits over 50 and 125 years, followed by instantaneous change in $U$ to 0.45

**Comment [cwt9]:** I've changed this text and the text of figure 3 to try and clarify Anon's identification of confusion over model run times, etc.

**4 Results**

**4.1 Changes in morphology under gradual wave climate change**

[revised manuscript text omitted]

The results reported here provide *guidelines* for critical timescales of wave-climate change below which either capes or spits would fail to respond in a quasi-equilibrium fashion. These set limits on the mode of coastline adjustment that will occur: when wave climate changes occur over timescales shorter than the characteristic timescale for morphological adjustment, the shoreline response will exhibit significant morphologic memory. In contrast, longer timescales for change permit a quasi-equilibrium response, in which morphological adjustment keeps pace with the change in wave climate. This scaling makes it possible to

**Comment [cwt15]:** Anon's comment 7 seems somewhat superfluous here. We don't say that all coastlines are likely to be out of equilibrium with current forcing, only that one should not assume quasi- or dynamic equilibrium; there can be situations where coastlines are out of equilibrium. We think this is implicitly clear. But we've amended the text slightly.

**Comment [cwt16]:** Itacilicised for emphasis…

[revised manuscript text omitted]

Comment [cwt20]: This figure has been revised hopefully to clarify. The text in the ms has been augmented and amended.

**Figure S1**

The metrics for capes used in the calculations of the diffusive scaling relationship between wavelength and time. The figure accompanies Table S1.

**Figure S2**

The metrics for spits used in the calculations of the diffusive scaling relationship between wavelength and time. The figure accompanies Table S2.

Figure 1

[Figure]

[Figure]

Capes, North Carolina coast, USA

[Figure]

Flying spits, Namibia          Low-angle connected spits, Norfolk, UK

Figure 2

(a)

[Figure]

(b)

[Figure]

a: CEM model scheme
b: Wave climate probability density functions
c: Wave climate phase space

Figure 3

[Figure]

**Figure 4**

**Capes; *A* = 0.5**

**(a.i)**

Aspect ratio vs. Time, years

**(a.ii)**

Wavelength, km vs. Time, years

**(a.iii)**

Amplitude, m vs. Time, years

**Spits; *A* = 0.7**

**(b.i)**

Aspect ratio vs. Time, years

**(b.ii)**

Wavelength, km vs. Time, years

**(b.iii)**

Amplitude, m vs. Time, years

[Figure]

Static ***U*** = 0.55     ***U*** = 0.7 to 0.55     ***U*** = 0.7 to 0.45

— — — — — Time interval over which wave climate is changed

e-folding times for changes in aspect ratio for ***U*** changing *instantaneously* from 0.7 to 0.45 (see Figure 5)

— — — — — 1st e-folding time for 900 year models: capes: 20 years; spits: 90 years

— — — — — 2nd e-folding time for 900 year models: capes: 80 years; spits: 320 years

Figure 5

[Figure]

Capes; **A** = 0.5            Spits; **A** = 0.7

(a.i)            (b.i)

(a.ii)            (b.ii)

(a.iii)            (b.iii)

———— Static **U** = 0.55      ———— **U** = 0.7 to 0.55      ———— **U** = 0.7 to 0.45

– – – – – Time at which wave climate is changed instantaneously

e-folding times for changes in aspect ratio for **U** changing *instantaneously* from 0.7 to 0.45

– – – – – 1st e-folding time for 900 year models: capes: 20 years; spits: 90 years

– – – – – 2nd e-folding time for 900 year models: capes: 80 years; spits: 320 years

Figure 6

[Figure]